# Concordance of the Histopathologic Diagnosis of Concurrent Duodenal and Ileal Biopsy Specimens in Dogs

**DOI:** 10.3390/ani11102938

**Published:** 2021-10-11

**Authors:** Sarah Caulfield, Simon L. Priestnall, Aarti Kathrani

**Affiliations:** 1Small Animal Internal Medicine Service, Lumbry Park Veterinary Specialists, CVS Group Plc, Hampshire GU34 3HL, UK; sarah.caulfield@cvsvets.com; 2Department of Pathobiology and Population Sciences, Royal Veterinary College, Hertfordshire AL9 7TA, UK; spriestnall@rvc.ac.uk; 3Department of Clinical Science and Services, Royal Veterinary College, Hertfordshire AL9 7TA, UK

**Keywords:** histopathology, endoscopy, full-thickness, duodenum, ileum, lymphoma

## Abstract

**Simple Summary:**

Canine chronic inflammatory enteropathy (CIE) is a gastrointestinal (GI) inflammatory condition that requires histopathology of intestinal biopsies for definitive diagnosis. This can be performed endoscopically or surgically; with the former being more common as it is less invasive. In recent years, research has identified that different types of inflammatory disease and histologic severity can exist at different locations within the GI tract, known as discordance. There is a lack of understanding of the clinical impact of lesion oversight in dogs with discordant biopsies between locations. Therefore, current recommendations are to procure endoscopic biopsies from both the upper intestinal (stomach and duodenum) and lower small intestinal tract (ileum) when possible. The rate of discordance between different GI locations has not been assessed in dogs with non-inflammatory enteropathies, i.e., neoplastic etiology, nor compared between endoscopic and surgically collected biopsy specimens. Here, we report that the rate of discordance between the duodenum and ileum in CIE is similar between endoscopic biopsy and surgical biopsy specimens and confirm that discordance between upper and lower GI biopsies exist for small-cell lymphoma. There were no clinicopathologic variables that were correlated with discordance. We conclude that for all dogs with chronic GI signs, concurrent duodenal and ileal biopsies should be performed.

**Abstract:**

Histopathologic discordance between gastrointestinal (GI) locations in canine chronic inflammatory enteropathy (CIE) has prompted recommendations to biopsy both the duodenum and ileum, while further evaluation is required for non-CIE. We aimed to determine the concordance of histopathologic diagnosis between duodenal and ileal endoscopic or full-thickness biopsy specimens for all dogs with CIE and GI neoplasia and to assess the association between histopathologic discordance between GI locations with clinicopathologic variables. Seventy-nine dogs were eligible, with endoscopic (74) or full-thickness (5) biopsy specimens. Clinicopathological data were recorded for all dogs. Concordance of histopathologic diagnosis was retrospectively assessed for concurrent duodenal and ileal biopsy specimens by a single board-certified veterinary pathologist using the modified World Small Animal Veterinary Association (WSAVA) Gastrointestinal Standardization Group guidelines. Sixty-seven dogs were diagnosed with CIE and 5 with enteric-associated T-cell lymphoma-2 (EATL-2). Concordance of histologic diagnosis between duodenal and ileal sites was similar between endoscopic (73.0%) and full-thickness (80.0%) biopsy groups. For the CIE cases, lymphoplasmacytic enteritis had the highest concordance (73.0%) and eosinophilic enteritis the least (16.7%). Of the 5 neoplastic cases, 5/5 (100%) were present at the duodenum but only 3/5 (60%) in the ileum. No clinicopathologic variables demonstrated a statistically significant association with discordance. We conclude that the level of discordance necessitates concurrent biopsy of both duodenum and ileum in all dogs with chronic GI signs. The rate of EATL-2 was lower than rates reported for cats.

## 1. Introduction

Investigations into canine chronic enteropathies frequently progress to biopsy of the gastrointestinal (GI) tract. This is usually performed via endoscopy, as it is relatively non-invasive compared to full-thickness biopsy [1]. Disparities in disease process, termed histopathologic discordance, exist between different intestinal locations [2]. Studies on histopathologic discordance in canine chronic inflammatory enteropathy (CIE) have found a poor concordance rate of 27% in histopathologic diagnosis and of 50% in severity between duodenal and ileal endoscopic and full-thickness biopsies, with a higher pathology detection rate in the ileum [3,4]. This significant histopathologic discordance in CIE means location targeted biopsy risks overlooking intestinal lesions. Currently, the impact this could have on patient treatment and outcome is not understood; therefore, recommendations include concurrent sampling of duodenal and ileal sites. This necessitates multi-site biopsy collection, which increases procedural risk. In the case of GI neoplasia, such as lymphoma, the impact of missing lesions is clear, with reports of poor long-term prognosis [5,6].

Gastrointestinal lymphoma accounts for 5–7% of GI neoplasms in dogs, but still occurs at lower rates than in cats where enteric associated T-cell lymphoma-2 (EATL-2), a small-cell lymphoma, occurs at rates as high as 26% amongst a population of cats with generalized GI signs [7,8]. In canines, enteric associated T-cell lymphoma-1 (EATL-1), a large cell lymphoma, is more common than EATL-2. Enteric associated T-cell lymphoma-2 carries a more favorable prognosis with a reported median survival time of 628 days in dogs receiving treatment, but reduced detection may be seen due to difficulties distinguishing this from lymphoplasmacytic infiltration [9,10,11,12]. With the increased use of immunohistochemistry (IHC) and PCR for antigen receptor rearrangement (PARR), improving our EATL-2 diagnostic capability, it is likely that this diagnosis in dogs may increase [9]. In feline EATL-2, predilection sites have been suggested to be the ileum and jejunum; however, evidence is conflicting with some studies identifying the majority of EATL-2 cases within the duodenum and jejunum [10,13,14,15]. This indicates the existence of histopathological discordance and has led to recommendations to sample these locations if neoplasia is suspected. A recent study in cats demonstrated that when IHC and PARR were combined with H&E assessment, the number of intestinal lymphoma cases diagnosed increased and, in contrast with previous studies, samples from the ileum rarely altered the diagnosis that was achieved from duodenal biopsy samples alone [16]. For canine EATL-2, the small intestine is the favored anatomical location and studies have similarly demonstrated cases where EATL-2 has only been detected within the ileum, but this has not been firmly established as a predilection site [17,18]. Conflicting evidence exists as to whether increased perforation risk is seen in dogs with ulcerated or neoplastic intestinal disease [19,20]. If a high level of histopathologic diagnosis discordance is seen, with canine neoplastic cases favouring the ileum, including cases assessed via IHC and/or PARR analysis then this may provide evidence for the need to biopsy the thinner-walled ileum.

The rate of concordance of histopathologic diagnosis has yet to be assessed amongst a generalized canine GI case population—for example, cases including inflammatory and neoplastic causes. Additionally, the correlation of clinicopathologic data with histopathologic discordance and therefore their value as indicators of discordance has not yet been established. Full-thickness biopsy samples have traditionally been considered to be of more adequate quality than endoscopic biopsy samples; however, the latter technique is less invasive, allows earlier initiation of treatment and also facilitates direct visualization of the intestinal mucosa [21]. Small intestinal lesions in cases of CIE typically occur diffusely within the intestinal mucosa, and unpublished data produced by Marsilio et al. has demonstrated that the total mucosal surface procured by full-thickness biopsy is comparatively less than that obtained via endoscopic biopsy techniques [22,23]. The impact of biopsy type on histopathologic concordance is not yet known.

The aim of our study was to determine the rate of concordance of histopathologic diagnosis between duodenal and ileal biopsy specimens in all dogs with chronic GI signs due to inflammatory or neoplastic disease. Secondly, we compared this concordance rate between full-thickness and endoscopic biopsy specimens. Finally, we aimed to determine if there was an association between histopathologic discordance and signalment, clinicopathologic, or ultrasonographic data.

## 2. Materials and Methods

### 2.1. Study Design

Electronic records from our referral teaching hospital (Queen Mother Hospital for Animals, Royal Veterinary College (RVC)) and the pathology laboratory database from the RVC from April 2008 to December 2019 were reviewed. The database was searched for all canine cases that had both duodenal and ileal biopsies concurrently performed via endoscopic or full-thickness (surgical) methods. This was a referral level case population.

### 2.2. Retrospective Study Criteria for Case Selection

Inclusion criteria included those dogs with chronic GI signs of at least 3 weeks’ duration that were ultimately diagnosed with CIE or GI neoplasia following concurrent biopsy via endoscopy or exploratory laparotomy of the upper small intestine and lower small intestine, referred to as the duodenum and the ileum, respectively, throughout this manuscript. Dogs diagnosed with CIE had to have appropriate diagnostic investigations to rule out other causes for inclusion into the study. All dogs with CIE had a complete blood count (CBC), serum biochemistry, fecal parasitology or empirical deworming, and abdominal ultrasound. Further diagnostic evaluation for the CIE cases also included additional procedures as indicated by the history, physical examination, CBC, serum biochemistry, and abdominal ultrasound examination findings: serum cobalamin concentration in 71 dogs [95.9%], serum folate concentration in 69 dogs [93.2%], pancreatic function testing (canine pancreatic lipase immunoreactivity in 59 dogs [79.7%] and trypsin-like immunoreactivity in 37 [50.0%]), basal cortisol concentration or ACTH stimulation test in 64 dogs (86.5%), pre- or pre- and post-prandial bile acid concentrations in 3 dogs (4.05%), and urinalysis in 69 dogs (93.2%), including for all dogs with hypoalbuminemia. Basal cortisol concentration was performed if findings on analysis of CBC (e.g., lymphocytosis, eosinophilia) or electrolytes (hyperkalemia, hyponatremia, hypercalcemia) raised suspicion of hypoadrenocorticism. An ACTH stimulation test was performed in cases where basal cortisol was <55 nmol/L [24]. A bile acid stimulation assay was performed in cases where findings on biochemistry (decreased urea, cholesterol or glucose concentration) or diagnostic imaging (decreased hepatic size) were indicative of hepatic dysfunction. Cases were excluded if results were compatible with hypoadrenocorticism or hepatic dysfunction, the latter according to laboratory established reference ranges.

Twenty-one dogs were excluded from the study due to GI signs of less than 3 weeks’ duration and 16 were excluded due to incomplete diagnostic investigations for CIE, as indicated by the history, physical examination, CBC, serum biochemistry and abdominal ultrasound (9) or due to significant concurrent disease (7). Eight dogs were excluded due to lack of signalment and clinical historical data.

### 2.3. Histopathologic Grading and Review of Biopsy Specimens

An updated version of the peer-reviewed World Small Animal Veterinary Association (WSAVA) gastrointestinal histopathology scoring system, created by the WSAVA GI Standardization Group, was used for section interpretation and standardization for both biopsy sites to assess histologic diagnosis [1,25]. Four histologic parameters representing inflammation were assessed and recorded; lamina propria lymphocytes, lamina propria plasma cells, lamina propria eosinophils and lamina propria neutrophils. Where inflammatory cell infiltrate was identified and considered to be above normal, as per the updated WSAVA scoring system, severity was assessed as a singular score from grading of lamina propria infiltrate using a four-point scale: 0—normal, 1—mild, 2—moderate, 3—severe/marked. Biopsy results were classed as ‘mixed enteritis’ where densities of each cell type were increased above normal across lymphocytes, plasma cells, neutrophils and eosinophils within the same biopsy specimen. Diagnosis of lymphoma was made based upon a monomorphic infiltrate of lymphocytes, with notable paucity of plasma cells and/or evidence of significant epitheliotropism [1]. Morphological features were not assessed beyond lacteal dilation where a histologic diagnosis of lymphangiectasia was made.

### 2.4. Assessment of Concordance and Discordance

Each case was evaluated for cellular infiltrate concordance and was classed as concordant where the inflammatory or neoplastic cell type was reciprocal between the duodenal and ileal sites. Concordance of histologic severity was only assessed in those cases demonstrating concordance in inflammatory cell infiltrate at the duodenal and ileal sites. Severity scores were considered concordant if the same or adjacent severity scores were noted at both sites, except in the case of normal infiltration, which was classed as discordant if the histologic score at the second site was anything other than normal. For example; mild lymphoplasmacytic infiltrate in the duodenum was classed as concordant if mild or moderate lymphoplasmacytic infiltrate was found in the ileum; however, this was classified as discordant if severe lymphoplasmacytic infiltrate was noted within the ileum. This is to account for the subjectivity of the severity score, as significant inter-observer variation is seen even when using the WSAVA GI standardization group criteria [26].

To standardize evaluation, a single board-certified veterinary pathologist with a specialist interest in gastrointestinal pathology reviewed all suitable cases. At this stage, cases were excluded if the quality of the histopathologic sections was deemed inadequate or marginal due to incomplete length of villi and depth of crypts or if slides were no longer available for review. Slides from duodenal and ileal sections were reviewed concurrently for each case and the pathologist was blinded to the clinical background.

### 2.5. Signalment, Clinicopathologic and Ultrasonographic Variables

Signalment (age, breed, sex, body weight and neuter status), clinicopathologic variables (serum albumin, globulin, cobalamin and folate concentrations) and ultrasonographic abnormalities were recorded for each case. Clinicopathologic variables for each case included whether hypoalbuminemia (<28 g/L), hypocobalaminemia (<200 ng/L), hypoglobulinemia (<21 g/L) and hypofolatemia (<7.1 μg/L) were present. Ultrasonographic abnormalities were classified as mesenteric lymphadenopathy, intestinal mucosal speckling, thickening of the intestinal mucosa, focal peritoneal effusion, and loss of intestinal wall layering.

### 2.6. Immunohistochemistry and PARR Analysis

Further analysis was performed in three cases at the time of diagnosis where the hematoxylin and eosin (H&E) histopathologic diagnosis had been classed as inflammatory but concern was held by the pathologist that the cellular infiltrate was predominantly lymphocytic. Two cases underwent immunohistochemistry (IHC) and one underwent both IHC and PARR analysis, confirming the presence of lymphoma.

### 2.7. Ethical Considerations

The Ethics and Welfare Committee at the RVC granted ethical approval for this study (URN SR2019-0493).

### 2.8. Data Analysis and Statistics

Median and interquartile range were calculated for age, body weight, and body condition score. Binary logistic regression analysis was conducted to assess for significant predictive associations between histopathologic diagnosis discordance and signalment (1. age, categorized into four categories—0–24 months, 25–72 months, 73–120 months, and 121–168 months; 2. sex and neuter status, categorized into 4 categories—male, male neutered, female, and female neutered), clinicopathologic (1. albumin, categorized as presence or absence of hypoalbuminemia (<28 g/L); 2. globulin, categorized as presence or absence of hypoglobulinemia (<21 g/L); 3. cobalamin, categorized as presence or absence of hypocobalaminemia (<200 ng/L); and 4. folate, categorized as presence or absence of hypofolatemia (<7.1 μg/L)), ultrasonographic findings (categorized as unremarkable, mesenteric lymphadenopathy, intestinal mucosal speckling, thickening of the intestinal mucosa, focal peritoneal effusion or loss of intestinal wall layering), and biopsy method (categorized as endoscopy versus full-thickness). Variables associated with histologic diagnosis discordance with *p* < 0.20 in binary logistic regression were entered into multivariable analyses. In the multivariable regression models, analyses were performed in a backward stepwise manner. All variables with *p* < 0.20 were initially included, and the variable with the highest *p*-value was removed until all remaining variables had a *p* < 0.05. This was performed using the IBM SPSS (Statistical Product and Service Solutions, IBM, NY, USA) version 26 statistical software program.

Agreement of histopathologic diagnosis was assessed by κ analysis. Agreement was evaluated for all histopathologic diagnoses and then individually for lymphoplasmacytic inflammatory infiltrate and lymphoma. Interpretation of κ values was as follows: <0 no agreement, <0.20 slight agreement, 0.21–0.40 fair agreement, 0.41–0.60 moderate agreement, 0.61–0.80 substantial agreement, 0.81–1.00 almost perfect agreement [3,4]. Statistical significance was placed at *p* < 0.05.

## 3. Results

### 3.1. Dogs

In total, 79 dogs were included in the study. Seventy-four underwent endoscopic biopsies and 5 underwent full-thickness biopsies.

There were 41 females (35 female neutered) and 33 males (20 male neutered) in the endoscopic biopsy group. The full-thickness biopsy group comprised 2 females (all female neutered) and 3 males (all male neutered). The breeds within the endoscopic biopsy group included German Shepherd (10), Labrador (7), crossbreed (6), Rottweiler (5), Cocker Spaniel (4), and Staffordshire Bull Terrier (4). The breeds within the full-thickness biopsy group included; Dalmatian (1), Chihuahua (1), cross breed (1), Great Dane (1), and Pug (1). The median age of the biopsy groups were as follows: 49.0 months (interquartile range (IQ) 10.0–158.0 months) for the endoscopic biopsy group and 81.0 months (IQ 24.0–144.0 months) for the full-thickness biopsy group. Median bodyweight was 24.4 kg (IQ 3.9–54.1 kg) for the endoscopic biopsy group and 10.5 kg (IQ 5.5–63.5 kg) for the full-thickness biopsy group. The median body condition score was 4/9 (IQ 2–7/9) for the endoscopic biopsy group and 3.5/9 (IQ 2.5–7/9) for the full-thickness biopsy group. Data for body condition score were not available for 6/74 endoscopic biopsy cases.

### 3.2. Gastrointestinal Clinical Signs

In the endoscopic biopsy group, clinical signs were as follows: diarrhea (69 dogs, characterized as large bowel 15, small bowel 15 and mixed bowel 39, hemorrhagic diarrhea (6), vomiting (35 dogs, of which 6 where hemorrhagic), lethargy (15), weight loss (31), reduced appetite (24), coprophagia (4), abdominal pain (4), melena (3), regurgitation (1), tenesmus (1), increased appetite (1), bloat (1), and borborygmi (1).

In the full-thickness biopsy group, clinical signs included mixed bowel diarrhea (3), vomiting (4), regurgitation (2), weight loss (3), decreased appetite (2), peripheral lymphadenopathy (1), and bloat (1).

### 3.3. Clinicopathologic Variables

In the endoscopic biopsy group, data were missing for cobalamin in one case and for folate in four cases. Hypoalbuminemia (<28 g/L) was recorded in 27 of 74 cases, hypoglobulinemia (<21 g/L) was detected in 17 of 74 cases, hypocobalaminemia (<200 ng/L) was noted in 33 of 73 cases, and hypofolatemia (<7.1 μg/L) was identified in 18 of 70 cases.

In the full-thickness biopsy group, data for albumin and globulin was available for all cases but absent for cobalamin in 2 cases and for folate in 2 cases. Hypoalbuminemia (<28 g/L) was seen in 2 of 5 cases, hypocobalaminemia (<200 ng/L) was noted in 2 of 3 cases, and hypofolatemia (<7.1 μg/L) was identified in 1 of 3 cases.

### 3.4. Ultrasonographic Abnormalities

In the endoscopic biopsy group, 17 of the 74 cases had abnormalities detected on abdominal ultrasound. Abnormalities identified were mesenteric lymphadenopathy (9), intestinal mucosal speckling (5), thickening of the intestinal mucosa (2), and loss of intestinal wall layering (1).

In the full-thickness biopsy group, 2 of 5 cases had abnormalities detected on abdominal ultrasound. Abnormalities noted were mesenteric lymphadenopathy (1) and thickening of the intestinal mucosa (1).

### 3.5. Final Diagnosis

In the endoscopic biopsy group, histopathologic diagnosis was of chronic inflammatory disease in 64 of 74 cases and EATL-2 in 5 of 74 cases. In 5 cases that were originally diagnosed as CIE based on mild inflammatory infiltrate, histopathologic review for the purposes of this study deemed both the duodenal and ileal biopsy specimens to have a normal infiltrate (Table 1).

In the full-thickness biopsy group, histopathologic diagnosis was of chronic inflammatory disease in 3 of 5 cases. For the remaining 2 cases, both were originally diagnosed as CIE based on mild inflammatory infiltrate; however, histopathologic review for the purposes of this study deemed both the duodenal and ileal biopsy specimens to have normal infiltrate. There were no neoplastic cases (Table 2).

The final histopathologic diagnosis and inflammatory scores for those cases with chronic inflammatory enteropathy within the endoscopic biopsy and full-thickness biopsy groups is depicted, irrespective of concordance or discordance between the duodenal and ileal sites, in Table 3.

### 3.6. Concordance and Discordance

For the endoscopic biopsy group, moderate agreement between histopathologic diagnoses was found (κ = 0.578). Fifty-four of the 74 cases (73.0%) demonstrated concordance in both inflammatory and neoplastic cell type. Twenty of the 74 cases (27.0%) showed discordance at the duodenal and ileal sites (Table 4). The highest rates of concordance existed for the histopathologic diagnosis of lymphoplasmacytic enteritis (27 of 37 total cases (κ = 0.725, substantial agreement)), lymphoplasmacytic and neutrophilic enteritis (4 of 6 cases (κ = 0.786, substantial agreement)), and lymphoma (3 of 5 cases (κ = 0.737, substantial agreement)).

In 4 of the endoscopic biopsy cases, the duodenum was found to be histologically normal in conjunction with inflammatory disease at the ileum. In 7 cases, the ileum was found to be normal with duodenal inflammatory disease identified.

In the full-thickness biopsy group, substantial agreement was noted (κ = 0.737). Four of the 5 cases (80%) demonstrated concordance in inflammatory cell type. One of the 5 cases (20%) showed discordance at the duodenal and ileal sites (Table 4).

Discordance of severity was shown in 6 of 74 cases (8.2%) within the endoscopy biopsy group and 1 of 5 cases (20%) within the full-thickness biopsy group.

### 3.7. Neoplasia

Neoplasia was solely diagnosed in the endoscopic biopsy group, with 5 of the 74 endoscopy specimens demonstrating EATL-2. Three cases in the endoscopic biopsy group of EATL-2 were diagnosed in both the duodenal and ileal biopsy specimens. Of these, 2 cases presented with hypoalbuminemia (<28 g/L), 1 with hypocobalaminemia (<200 ng/L) and 2 with hypofolatemia (<7.1 μg/L). Two had ultrasonographic abnormalities identified, 1 presented with intestinal mucosal speckling and 1 with mesenteric lymphadenopathy. Two EATL-2 cases were diagnosed solely within the duodenum with moderate lymphoplasmacytic and eosinophilic inflammatory disease diagnosed within the ileum. These cases demonstrated hypoalbuminemia (<28 g/L) and hypocobalaminemia (<200 ng/L).

### 3.8. Immunohistochemistry and PARR Analysis at Original Diagnosis

Three of the endoscopic biopsy cases underwent further testing at the time of original histopathologic diagnosis of marked lymphoplasmacytic inflammation: IHC (CD3 and CD20) alone was performed in 2 cases; CD3 diagnosed positive EATL-2 in both the duodenum and ileum in 1 case and EATL-2 located solely within the duodenum in the other. An additional case diagnosed as marked lymphoplasmacytic inflammation underwent IHC (strongly CD3 positive, weakly CD79 positive) and PARR analysis (B clonal lymphocyte population) and was found to be consistent with lymphoma (EATL-2) in both the duodenal and ileal locations. One case of both duodenal and ileal EATL-2 and 1 duodenal EATL-2 case were diagnosed based on H&E assessment alone.

### 3.9. Statistical Analysis of Signalment, Clinicopathologic and Ultrasound Variables

No variables were found to have a statistically significant association with discordance of histopathologic diagnosis in the univariable analysis: biopsy type (endoscopic biopsy versus full-thickness biopsy) *p* = 0.57, age *p* = 0.70, sex and neuter status *p* = 0.88, albumin *p* = 0.23, globulin *p* = 0.43, cobalamin *p* = 0.53, folate *p* = 0.22, and ultrasound findings *p* = 0.73. Multivariable analysis was not performed as no variable had a *p* < 0.20.

## 4. Discussion

This is the first study assessing the concordance of histopathologic diagnosis between duodenal and ileal biopsies in dogs with both inflammatory and neoplastic etiologies and contrasting concordance rates between endoscopic biopsy and full-thickness biopsy. In addition, we evaluated whether predictive indicators exist within signalment, clinicopathologic or ultrasonographic variables for discordance of histopathologic diagnosis between the two sites.

Our study showed that a similar level of concordance in histopathologic disease diagnosis exists between endoscopic and full-thickness duodenal and ileal biopsy specimens, with moderate to substantial agreement identified, respectively. This was higher than the 27% agreement reported by Casamian-Sorrosal et al. in dogs with canine chronic enteropathy, although none of their cases were diagnosed with neoplasia [3]. We found higher rates of concordance when agreement between severity of disease was not evaluated, which is supported by findings from the literature [3,4]. The discordance between duodenal and ileal sites was represented by an inflammatory or neoplastic disease process and in some cases a normal level of cellular infiltration. In 15.0% of our endoscopic biopsy cases, one of the two intestinal sites was found to be histologically normal with significant inflammatory infiltrate identified at the other site. This is similar to findings by Procoli et al., who identified this in 21.0% of their canine chronic enteropathy cases [4]. Following further review, a higher proportion of ileal biopsy samples with a normal population of inflammatory cells was paired with histologically diseased duodenal biopsy samples in our population. This contrasts with other studies where more diseased ileal tissue was found in conjunction with unaffected duodenal tissue [3,4]. This could be due to the duodenum being more accessible, which facilitates sampling along its length and increases the likelihood of obtaining affected tissue. This is compared to the ileum where access is more difficult and so biopsies of a smaller region may be obtained. Additionally, a blind biopsy technique can be performed when access to the ileum cannot be achieved via endoscopy, which again increases the risk of missing lesions, especially if the ileal mucosa cannot be visually inspected. This suggests that when investigating gastrointestinal cases where inflammatory or neoplastic disease is suspected, it is most important to obtain high quality duodenal biopsies. Unfortunately, outcome data were not available for our cases, therefore we could not evaluate the impact of histopathologic discordance on our CIE cases.

Numerous studies have failed to demonstrate associations between histopathologic cell type and severity with the degree of clinical disease, treatment and outcome [27,28,29,30]. It has been proposed that the absence of an association relates to inappropriate patient selection for biopsy and the lack of applicability of scoring systems, such as the WSAVA guidelines, to a real-world patient population [25,26]. Until this is resolved, the relevance of this discordance for clinical care remains to be clarified. Additionally, no variables were identified as predictive factors of discordance of histopathologic diagnosis. Therefore, our study cannot confirm whether duodenal and ileal samples are needed based on the case presentation and prior investigative findings alone. The lack of association between clinicopathologic variables and the level of histopathologic discordance between duodenum and ileum in our study, and the lack of understanding regarding the clinical impact of such histopathologic discordance, suggests that concurrent duodenal and ileal biopsy samples should be ideally obtained where possible.

In seven of the CIE cases from this case population (five from the endoscopy biopsy group and two from the full-thickness biopsy group) a review of the slides for the purposes of this study altered their diagnosis from mild CIE to normal biopsy infiltrate. These slides were taken from older specimens within the biopsy pool and since the initial reporting the following guidelines have been produced; WSAVA Guidelines for evaluation of gastrointestinal inflammation in companion animals 2010 and the simplified scoring system for defining gastrointestinal inflammation produced by Allenspach et al. in 2019 [1,25]. The use of these systems has improved the definition of histopathologic parameters and their grading, allowing more confidence in interpreting ‘normal’ biopsy infiltration.

Within our case population, the rate of EATL-2 was 6.3%. This rate of lymphoma is comparable to other studies, where an overall T-cell lymphoma rate (EATL-1 and EATL-2) of 8.0% was identified in full-thickness biopsies taken from dogs [2]; unusually, all lymphoma cases were of EATL-2 origin in our case population. The lymphoma cases within our study demonstrated substantial agreement, as 60.0% of the EATL-2 cases occurred concurrently at both duodenal and ileal sites. The H&E histopathologic review for 3 of these cases identified marked lymphoplasmacytic enteritis prior to further analysis. A similar level of substantial agreement was seen for lymphoplasmacytic enteritis cases within our case population. There is concern that lymphoplasmacytic inflammation may evolve into lymphoma and these results suggest that monitoring cases with concordant lymphoplasmacytic enteritis diagnoses for refractory signs is important [5,31]. In our study, IHC and PARR analysis were only performed in those cases that aroused suspicion of underlying EATL-2; i.e., cases with marked lymphocytic infiltrate, therefore it is possible that not all EATL-2 cases were captured based on H&E assessment alone. The predilection of EATL-2 cases in our study was for the duodenum, which was similar to a recent study where 85% of EATL-2 cases occurred at the duodenum compared to only 54% in the ileum [12]. This is in contrast to a number of studies on feline EATL-2 where the jejunum and ileum have been reported as favored locations; however, another study found that the majority of cases occurred in the duodenum and jejunum [10,13,14,15,32]. This has led to the suggestion of full-thickness biopsy as the best method in these cases due to the difficulty of sampling the jejunum and ileum via endoscopy. Forty percent of the EATL-2 cases in our study occurred in the duodenum alone with lymphoplasmacytic and eosinophilic inflammatory disease identified in the ileum. In those cases assessed via IHC or IHC and PARR, EATL-2 was detected in both the duodenum and ileum or in the duodenum alone, and in none of these cases would sampling of the ileum have altered the diagnosis already obtained in sampling the duodenum. This is similar to recent findings in feline intestinal lymphoma, where sampling of the ileum rarely changed diagnosis when biopsies were assessed with a combination of H&E staining, IHC and PARR [16]. Clinically, this could lead to prioritization of upper GI endoscopy where EATL-2 is suspected, but another study has found EATL-2 occurring at the ileum in many dogs, including detection in the ileum alone [17]. Given the differences between the data groups and the low number of neoplasia cases within our study, it is difficult to draw definitive inferences in EATL-2 incidence and predilection sites from our study alone. Interestingly, the neoplasia cases in our study existed solely within the endoscopic biopsy group. Whilst this could, in large part, be due to the larger number of cases within this biopsy group, feline EATL-2 has been shown to manifest initially in the mucosal layer for which endoscopic biopsy sampling may be better suited than full-thickness biopsy [13,14,15]. Full-thickness biopsy has been demonstrated to provide comparatively smaller samples of mucosa and a lower number of samples at each site [23]. With evidence demonstrating that ileal biopsy rarely impacted diagnosis in cases of feline EATL-2 and the possibility of reaching the proximal jejunum in small dogs with gastroduodenoscopy, this calls into question the need to use full-thickness biopsy in these cases unless a lesion has been noted at the mid jejunum [16,21]. Further research is required on the need for concurrent ileal biopsy in dogs with suspected EATL-2 if IHC and PARR analysis are also used before any definitive conclusions regarding the requirement for duodenal and ileal biopsy can be made.

Whilst no statistically significant clinicopathologic variables were predictive of histopathologic discordance, hypoalbuminemia and hypocobalaminaemia were found in 80 percent of EATL-2 cases. This is consistent with findings in other studies where 50.0–69.0% of EATL-2 cases had hypoalbuminemia and 40.0–71.0% of cases had hypocobalaminemia [12,17]. This confirms the importance of considering EATL-2 as a differential diagnosis in cases with low serum albumin and cobalamin and both have been highlighted as negative prognostic indicators for the survival of dogs with CIE [27]. Two EATL-2 cases in our study were diagnosed on H&E histopathology alone; three EATL-2 cases further underwent IHC or a combination of IHC and PARR. This supports the diagnostic capability of H&E histopathology alone for EATL-2, but also confirms that IHC and PARR are often required to differentiate inflammatory disease from EATL-2 in both dogs and cats, with as many as 60.0% of EATL-2 cases requiring further confirmation via IHC [9,17,33,34,35]. One of the EATL-2 endoscopic biopsy cases underwent IHC and was strongly CD3 positive, but also demonstrated a weak positivity for CD79a. Moreover, PARR analysis identified a B-clonal population. PARR lacks sensitivity for canine intestinal lymphoma with sensitivities of only 66.7–75% being reported, which is speculated to be due to the significant level of inflammatory infiltration seen alongside neoplastic cells in cases with EATL-2 [9,34]. Its specificity for EATL-2 has also been brought into question since 51% of CIE cases in a study on Shiba dogs demonstrated monoclonal populations [36]. CD79a is a B-cell marker, co-expression of CD3 and CD79a has been identified in canine nodal lymphoma and co-expression of CD3 and CD20 has been documented in canine EATL-1 [37,38]. Given the weak expression of CD79a in the case from our study, and based on H&E assessment alongside strong staining for CD3, this case was diagnosed with EATL-2; however, the presence of co-expression and potential lack of sensitivity and specificity of PARR for EATL-2 highlights the need for cases to be assessed with combined techniques using H&E, IHC and PARR analysis. We have identified this as an area for future research, particularly around the value of IHC and PARR analysis, potentially leading to greater detection of GI lymphoma and more specifically EATL-2 in dogs.

Our study had limitations in that the data group that underwent endoscopic biopsy (74) was larger than the full-thickness biopsy group (5 cases). We suspect this relates to a higher frequency of less invasive endoscopic biopsy being performed. Additionally, the location-targeted aspect of full-thickness biopsy meant that a number of full-thickness biopsy cases did not meet the inclusion criteria of concurrent duodenal and ileal biopsy. The endoscopic biopsy population included a higher proportion of cases with chronic GI signs and complete prior diagnostic investigation compared to the full-thickness biopsy population and as such this led to more exclusions being made within the full-thickness biopsy group. Our study did not assess morphologic criteria for chronic inflammatory enteropathies, beyond lacteal dilation for lymphangiectasia diagnosis, as it more specifically aimed to assess the concordance of inflammatory and neoplastic enteropathies. In addition, the concordance of various morphologic criteria of CIE between intestinal locations has been evaluated previously by Procoli et al. 2013 [4]. Our study cohort did not include any dogs diagnosed with EATL-1, the more common alimentary lymphoma in dogs. This is likely to have occurred due to cases with EATL-1 being more readily identified from ultrasonographic abnormalities and then diagnosed cytologically by fine needle aspirates or by histopathology of surgical biopsies that were lesion targeted and therefore did not match the criteria for case selection.

## 5. Conclusions

In conclusion, there were similar rates of histopathologic concordance and discordance for duodenal and ileal biopsies between endoscopic and full-thickness biopsy specimens. Canine EATL-2 demonstrated a predilection for the duodenum in our case population. Until we better understand the clinical impact of histopathologic discordance, concurrent biopsy of the duodenum and ileum may be ideal in cases suspected to involve CIE. Further research with larger case populations and an improved understanding of the clinical impact of histopathologic discordance between the duodenum and the ileum are needed.

## Figures and Tables

**Table 1 animals-11-02938-t001:** Final histopathologic diagnosis following re-review of slides within the endoscopic biopsy group of inflammatory or neoplastic disease (74 dogs). Concordant cases represented first, followed by discordant cases where individual corresponding diagnosis per site is indicated.

Histopathologic Diagnosis: Endoscopic Biopsy Group	Number of Cases with Concordant Diagnosis at Duodenal and Ileal Sites:	Discordance in Histopathologic Diagnosis by Site:	
		Duodenal Diagnosis:	Ileal Diagnosis:
Lymphoplasmacytic enteritis	27	5	5
Neutrophilic enteritis	1	1	2
Eosinophilic enteritis	1	2	3
Plasmacytic enteritis	4	5	0
Lymphoplasmacytic and eosinophilic enteritis	2	2	5
Lymphoplasmacytic and neutrophilic enteritis	4	2	0
True ‘mixed’ enteritis	1	1	2
Lymphangiectasia	2	0	0
Normal biopsy infiltrate	5	4	7
Lymphoma (EATL-2)	3	2	0

**Table 2 animals-11-02938-t002:** Final histopathologic diagnosis following re-review of slides within the full-thickness biopsy group of inflammatory or neoplastic disease (5 dogs). Concordant cases represented first, followed by discordant cases where individual corresponding diagnosis per site is indicated.

Histopathologic Diagnosis: Full-Thickness Biopsy Group	Number of Cases with Concordant Diagnosis at Duodenal and Ileal Sites:	Discordance in Histopathologic Diagnosis by Site:	
		Duodenal Diagnosis:	Ileal Diagnosis:
Lymphoplasmacytic enteritis	1	0	0
Neutrophilic enteritis	0	1	0
Lymphoplasmacytic and eosinophilic enteritis	1	0	0
Eosinophilic and neutrophilic enteritis	0	0	1
Normal biopsy infiltrate	2	0	0

**Table 3 animals-11-02938-t003:** Final histopathologic diagnosis at both the duodenal and ileal sites for chronic inflammatory enteropathy cases depicted with the inflammatory score. The numbers within the table represent the number of cases that had this final diagnosis at the individual location. This table depicts both endoscopic biopsy cases (74 dogs) and full-thickness biopsy cases (5 dogs). The key explains the definitions for each inflammatory score based upon inflammatory infiltrate type. Adapted from Allenspach et al., 2019 [25].

Key								
Duodenal and Ileal Histopathological Inflammatory Score	Normal = 0		Mild = 1		Moderate= 2		Severe/Marked = 3	
Lamina propria lymphocytes and plasma cells (% area of one 400× villous field or cells between crypts)	≤25, ≤2		26–50, 3–5		51–75, 6–10		≥76, ≥11	
Lamina propria eosinophils (cells per 400× field)	≤3		4–10		11–20		≥21	
Lamina propria neutrophils (cells per 400× field)	0		≤10		11–30		≥31	
**Endoscopic biopsy specimens:**								
**Histopathologic diagnosis (enteritis) (Number of cases):**	**Normal = 0**		**Mild = 1**		**Moderate = 2**		**Severe/marked = 3**	
	**Duodenum**	**Ileum**	**Duodenum**	**Ileum**	**Duodenum**	Ileum	**Duodenum**	**Ileum**
Lymphoplasmacytic			25	20	7	10	1	2
Neutrophilic			1	3	2	1		
Eosinophilic			2	1		2		
Plasmacytic			6	3	3	1		
Lymphoplasmacytic and eosinophilic			4	3	0	5		
Lymphoplasmacytic and neutrophilic			2	1	3	2		1
Mixed			1	1	1	1		
Normal mucosal infiltrate	10	14						
**Full-thickness biopsy specimens:**								
**Histopathologic diagnosis (enteritis) (Number of cases):**	**Normal = 0**		**Mild = 1**		**Moderate = 2**		**Severe/marked = 3**	
	**Duodenum**	**Ileum**	**Duodenum**	**Ileum**	**Duodenum**	**Ileum**	**Duodenum**	**Ileum**
Lymphoplasmacytic			1			1		
Neutrophilic			1					
Eosinophilic and neutrophilic				1				
Lymphoplasmacytic and eosinophilic			1	1				
Normal mucosal infiltrate	2	2						

**Table 4 animals-11-02938-t004:** Concordance and discordance of cell type between duodenal and ileal biopsy specimens for each biopsy type: full-thickness biopsy (5 dogs) versus endoscopic biopsy (74 dogs) with 95% confidence intervals (CI) reported.

Histopathology	Biopsy Type	
	Full-Thickness	Endoscopic
Concordance of cell type	4/5 cases	54/74 cases
	80.0%	73.0%
	95% CI 0.36–0.98	95% CI 0.62–0.82
Discordance of cell type	1/5 cases	20/74 cases
	20.0%	27.0%
	95% CI 0.02–0.64	95% CI 0.18–0.38

## Data Availability

Data are contained within the article. The data presented in this study are available in the tables in the article *Concordance of the Histopathologic Diagnosis of Concurrent Duodenal and Ileal Biopsy Specimens in Dogs*.

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
