# Peer review of "Concordance of the Histopathologic Diagnosis of Concurrent Duodenal and Ileal Biopsy Specimens in Dogs"

_animals, 2021, doi:10.3390/ani11102938_

Round 1
Reviewer 1 Report
Concordance of the histopathologic diagnosis of concurrent duodenal and ileal biopsy specimens in dogs is an interesting study/manuscript. I have a few comments/questions.
2.5 Signalment, clinicopathologic and ultrasonographic variables
Why have not you included muscularis propria (of the small intestine) thickening as one of the ultrasonographic abnormalities? You might consider adding this piece of information since thickening of the muscularis propria is associated with diffuse infiltrative bowel disease such as lymphoma or inflammatory bowel disease (at least in cats). And it seems to me that we can see this phenomenon more and more in dogs too (it might be because we pay more attention to it and use better and better US machines).
3.9 Statistical analysis of signalment, clinicopathologic and ultrasound variables
There is a typo (probably) here: ……., folate P=0.0.22, ………………..
It is interesting that all lymphoma cases in your group belong into EATL-2 category. We probably need to do more IHC and PARR at least on suspected severe lymphocytic inflammatory cases as you mentioned. Although the number of lymphoma cases in your study is relatively low. Also, it would be really interesting to know more about the therapy and progression of those cases (although those have not been the aims of the study).
Tables should be presented as stand-alone entities and therefore legends should specify the species and number of animals from which the data is derived.
Author Response
Reviewer: 1
The authors would like to thank this reviewer for the very constructive and helpful comments, which have improved the manuscript significantly.
Concordance of the histopathologic diagnosis of concurrent duodenal and ileal biopsy specimens in dogs is an interesting study/manuscript. I have a few comments/questions.
2.5 Signalment, clinicopathologic and ultrasonographic variables
Why have not you included muscularis propria (of the small intestine) thickening as one of the ultrasonographic abnormalities? You might consider adding this piece of information since thickening of the muscularis propria is associated with diffuse infiltrative bowel disease such as lymphoma or inflammatory bowel disease (at least in cats). And it seems to me that we can see this phenomenon more and more in dogs too (it might be because we pay more attention to it and use better and better US machines).
The authors thank the reviewer for this comment.
Reply: The authors considered using muscularis thickening as an ultrasonographic abnormality, however; this data set stretched over a large time period and muscularis thickening has not always been considered as a significant finding in dogs, therefore in an attempt to standardise the abnormalities that could have been detected via ultrasound and to do so in a robust manner this was not included. It is something that we would consider evaluating in a prospective study.
3.9 Statistical analysis of signalment, clinicopathologic and ultrasound variables
There is a typo (probably) here: ……., folate P=0.0.22, ………………..
The authors thank the reviewer for this comment.
Reply: This has now been altered, LINE 384, and now reads as “folate P= 0.22, and ultrasound…”
It is interesting that all lymphoma cases in your group belong into EATL-2 category. We probably need to do more IHC and PARR at least on suspected severe lymphocytic inflammatory cases as you mentioned. Although the number of lymphoma cases in your study is relatively low. Also, it would be really interesting to know more about the therapy and progression of those cases (although those have not been the aims of the study).
The authors would like to thank the reviewer for this comment.
Reply: The authors agree that this would be a very interesting avenue, whilst this was beyond the scope of this study, it is certainly something to be explored with future studies.
Tables should be presented as stand-alone entities and therefore legends should specify the species and number of animals from which the data is derived.
The authors would like to thank the reviewer for this comment.
Reply: This has now been altered within the legends of the tables in the manuscript.
Reviewer 2 Report
See comments in attached file

Author Response
Reviewer: 2
The authors would like to thank this reviewer for the very constructive and helpful comments, which have improved the manuscript significantly.
Simple Summary
Line 6-7 Wording is confusing. Consider rephrasing instead “There is a lack of understanding of the clinical impact of lesion oversight in dogs with discordant biopsies between locations, therefore current recommendations are to procure endoscopic biopsies from both the upper intestinal and lower small intestinal tract when possible.”
The authors thank the reviewer for this comment. LINE 14-18 now reads as follows:
‘There is a lack of understanding of the clinical impact of lesion oversight in dogs with discordant biopsies between locations, therefore current recommendations are to procure endoscopic biopsies from both the upper intestinal (stomach and duodenum) and lower small intestinal tract (ileum) when possible.’
Also, given the length of the canine duodenum / ileum and the length of most endoscopes, and the lack of endoscopic landmarks differentiating duodenum / ileum from jejunum, in small dogs one can likely obtain jejunal samples. Perhaps this should be mentioned somewhere in the paper and may be more appropriate to use upper and lower small intestines rather than specifying duodenum and ileum in this manuscript. If the authors wish to use “duodenum” and “ileum” throughout the manuscript, they should at least discuss the above points and say something along the lines of upper and lower small intestinal endoscopic biopsies are referred to as duodenum and ileum, respectively.
The authors thank the reviewer for this comment. The authors completely appreciate the reviewers point, we have elected to stick to referring to the upper small intestine as the duodenum and lower small intestine as the ileum, purely from the perspective that this fits with the histopathological reporting of these areas and to avoid confusion. However, please note that in order to make this clear LINE 133-135 has now been altered to read as follows:
‘Inclusion criteria included those dogs with chronic GI signs of at least 3 weeks duration that were ultimately diagnosed with CIE or GI neoplasia, following concurrent biopsy via endoscopy or exploratory laparotomy of the upper small intestine and lower small intestine, referred to as the duodenum and the ileum respectively throughout this manuscript.’
Line 9: Remove the space after the dash in “non- inflammatory”
The authors would like to thank the reviewer for this comment. LINE 22 now reads as follows:
‘GI locations has not been assessed in dogs with non-inflammatory enteropathies, i.e. neoplastic etiology…’
Line 9: Add a comma after “enteropathies” before “i.e.”
The authors would like to thank the reviewer for this comment. LINE 22 now reads as follows:
‘GI locations has not been assessed in dogs with non-inflammatory enteropathies, i.e. neoplastic etiology…’
Line 9-10: Add “etiology” after neoplastic, and comma after ; “infectious” should be removed as your study group had no GI infection so that is irrelevant to this paper ; replace “or” with “nor”
The authors would like to thank the reviewer for this comment. LINE 22-24 now reads as follows:
‘, i.e. neoplastic etiology, nor compared between endoscopic and surgically collected biopsy specimens.’
Line 10-11: Add “the rate of” before “discordance” ; add “biopsy” before “specimens”, and a comma after specimens.
The authors thank the reviewer for this comment. LINE 24 now reads as follows:
‘rate of discordance between the duodenum and ileum in CIE is similar between endoscopic biopsy and surgical biopsy specimens, and’
Line 12: Add “between upper and lower GI biopsies” after “discordance”
The authors would like to thank the reviewer for this comment. LINE 25-26 now reads as follows:
‘endoscopic biopsy and surgical biopsy specimens and confirm that discordance between upper and lower GI biopsies exists for small-cell lymphoma.’
Line 12: “small-cell lymphoma” hyphenated here, but not in other parts of the manuscript. Please keep it consistent.
The authors would like to thank the reviewer for this comment.
The manuscript has now been altered to read ‘small-cell cell lymphoma’ throughout the manuscript.
Line 13: Instead of “led to detection of”, consider “was correlated with”
The authors would like to thank the reviewer for this comment. Please note this has now been altered on LINE 27 to read; ‘There were no clinicopathologic variables that were correlated with discordance.’
Line 14: Comma after “signs”.
The authors would like to thank the reviewer for this comment. Please find that this has been altered on LINE 28 to read; ‘We conclude that for all dogs with chronic GI signs, concurrent duodenal and ileal biopsies should be performed.’
Abstract:
Line 26: Insert “the” before “duodenum”
The authors would like to thank the reviewer for this comment. Please find that this has been altered on LINE 30 to read; ‘has prompted recommendations to biopsy both the duodenum and…’.
Line 29: GI infection is irrelevant in this paper given your study population
The authors would like to thank the reviewer for this comment. Please find that this has been altered on LINE 33 to read; ‘all dogs with CIE and GI neoplasia, and to assess discordance with clinicopathologic’.
Line 29: Consider “…and to assess the association between histopathologic discordance between GI locations with clinicopathologic variables”.
The authors would like to thank the reviewer for this comment. Please find that this has been altered on LINE 33 to read; ‘and to assess the association between histopathologic discordance between GI locations with clinicopathologic variables’
Line 35: Again, GI infection is irrelevant in this paper given your study population
‘The authors would like to thank the reviewer for this comment. Please find that this has been altered on LINE 40 to read; ‘Seventy-four dogs were diagnosed with CIE and 5 with enteric-associated T-cell lymphoma-2 (EATL-2)’.
Introduction: (Line numbers are partly cut off from introduction onwards so I had to do my best with what I can deduce, I apologise if I make a mistake with some line numbers)
Line 54: Replace “alongside” with “with”
The authors would like to thank the reviewer for this comment. Please find that this has been altered on LINE 58 to read; ‘..full-thickness biopsies, with a higher pathology detection rate in the ileum..’.
Line 58-59: Consider “…collection, which increases procedural risk.”
The authors would like to thank the reviewer for this comment. Please find that this has been altered on LINE 63 to read; ‘This necessitates multi-site biopsy collection, which increases procedural risk’.
Line 59-60: The references cited focus more on large cell lymphoma / EATL-1, which do carry a poor prognosis. However, the prognosis for EATL-2 in dogs seems to be better in the Couto et al and Lane et al papers that you subsequently cited, and should be at least mentioned here, as that has relevance to the potential clinical impact in cases with histopathologic discordance, and all your LSA cases in your study are EATL-2
The authors would like to thank the reviewer for this comment. Please note we have altered the paragraph to read as follows: LINE 72.
‘Enteric associated T-cell lymphoma-2 carries a more favorable prognosis with reported mean survival times of 424-628 days in cases receiving treatment, but reduced detection may be seen due to difficulties distinguishing this from lymphoplasmacytic infiltration 9,10,11,12,19.’
Line 66: Comma after large cell lymphoma
The authors would like to thank the reviewer for this comment. Please find that this has been altered on LINE 71 to read; ‘…(EATL-1), a large cell lymphoma, is more common than EATL-2…’.
Line 72: In multiple other studies (Moore et al 2005, 2012, Carreras et al 2003, Chow et al 2021), EATL-2 in cats is found to be most prevalent in the upper small intestine (jejunum or duodenum). And the results of the Chow et al 2021 study suggests biopsies from the ileum / lower small intestine rarely changed the diagnosis. These references should be included and discussed if you are attempting to draw comparisons with cats.
The authors thank the reviewer for this comment. LINE 81-87 now reads as follows:
‘In feline EATL-2, predilection sites have been suggested to be the ileum and jejunum, however, evidence is conflicting with some studies identifying the majority of EATL-2 cases within the duodenum and jejunum, 10,13–15. This indicates the existence of histopathological discordance and has led to recommendations to sample these locations if neoplasia is suspected. A recent study in cats demonstrated that when IHC and PARR were combined with H&E assessment, the number of intestinal lymphoma cases diagnosed increased and in contrast with previous studies, samples from the ileum rarely altered the diagnosis that was achieved from duodenal biopsy samples alone 16.’
Lines 75-76: It would be more appropriate based on the results from the 2 cited references to say EALT-2 was detected the ileum in many dogs, including some dogs where it was only detected in the ileum, rather than using the term “predilection site” as, eyeballing the data, there were a large number of cases in both papers where it was detected in only the duodenum / jejunum as well.
The authors would like to thank the reviewer for this comment. Please find that this has been altered on LINE 87 to read; ‘For canine EATL-2, recommendations reflect those of CIE; the small intestine is the favored anatomical location and studies have similarly demonstrated cases where EATL-2 has only been detected within the ileum, but this has not been firmly established as a predilection site’.
Line 79, 83, 92, 109: All the mention of infectious causes are irrelevant in this manuscript
The authors thank the reviewer for this comment. Please find that this has been edited in all suggested locations:
Line 82: ‘If a high level of histopathologic diagnosis discordance is seen with canine neoplastic cases then this may provide evidence for the need to biopsy the thinner-walled ileum.’
Line 87: ‘The rate of concordance of histopathologic diagnosis has yet to be assessed amongst a generalized GI case population, for example, cases including inflammatory and neoplastic causes.’
Line 96: ‘The aim of our study was to determine the rate of concordance of histopathologic diagnosis between duodenal and ileal biopsy specimens in all dogs with chronic GI signs due to inflammatory or neoplastic disease.’
Line 86: Although historically full-thickness biopsies have been regarded as superior to endoscopically, most internists would argue that may not be true. Also, unpublished data by Marsilio (cited in Chow et al 2021) suggests the diagnostically available mucosal surface can be significantly less in full-thickness compared to endoscopic biopsies, and may be worth discussing. Furthermore, the advantages / disadvantages of full-thickness vs endoscopic biopsies should be briefly mentioned either here or later in the discussion.
The authors would like to thank the reviewer for this comment. The authors have now added additional information regarding the main advantages/disadvantages to this paragraph. LINE 102 now reads as follows:
‘Full-thickness biopsy samples have been considered to be of more adequate quality than endoscopic biopsy samples, however the latter technique is less invasive, allows earlier initiation of treatment and also facilitates direct visualization of the intestinal mucosa. 19 . Small intestinal lesions in cases of CIE typically occur diffusely within the intestinal mucosa, and unpublished data produced by Marilio et al, has demonstrated that the total mucosal surface procured by full-thickness biopsy is comparatively less than that obtained via endoscopic biopsy techniques 20,21. The impact of biopsy type on histopathologic concordance is not yet known.’
Line 88: Fullstop at the end of the sentence
The authors would like to thank the reviewer for this comment.
Line 111 now reads as follows: ‘Routinely, full-thickness biopsy samples have been considered to be of more adequate quality than endoscopic biopsy samples, but the impact of biopsy type on histopathologic concordance is not yet known.’
Line 95-96: Consider “…we aimed to determine if there was an association between histopathologic discordance and signalment, clinicopathologic, or ultrasonographic data.”
The authors would like to thank the reviewer for this comment.
Line 116 now reads as follows: ‘Finally, we aimed to determine if there was an association between histopathologic discordance and signalment, clinicopathologic, or ultrasonographic data.’
Line 103: Please state the specs of endoscope and biopsy forceps used. Also, please state (either here or in the results) if all endoscopic biopsies (especially ileal, or lower small intestinal biopsies) were obtained via direct visualization or if some / how many were blind biopsies. Although ileal biopsies can often be obtained with direct visualization, that isn’t always the case and readers may be interested to know how often the ileum was able to be cannulated vs blind biopsies in your study.
The authors thank the reviewer for this comment. Unfortunately, due to the retrospective nature of this study and the wide time span over which the study was conducted, it was not possible to record this information for all dogs when using the electronic medical records database.
Line 120: Why was hypoadrenocorticism not consistently ruled out?
The authors thank the reviewer for this comment. We have now added additional information to this paragraph, LINE 146-154 now reads as follows:
‘Basal cortisol concentration was performed if findings on analysis of CBC (e.g. lymphocytosis, eosinophilia) or electrolytes (hyperkalaemia, hyponatraemia, hypercalcaemia) raised suspicion of hypoadrenocorticism, an ACTH stimulation test was performed in cases were basal cortisol was <55nmol/l 22. A bile acid stimulation assay was performed in cases where findings on biochemistry (decreased urea, cholesterol or glucose concentration) or diagnostic imaging (decreased hepatic size) were indicative of hepatic dysfunction. Cases were excluded if results were compatible with hypoadrenocorticism or hepatic dysfunction, the latter according to laboratory established reference ranges.’
Section 2.5: Can you also include CCEAI score since you probably already have / can deduce that information, and assess the association with concordance / discordance as well if possible?
The authors would like to thank the reviewer for this comment. This was considered, however; this has been done previously by Procoli et al and as such the authors feel if this was to be assessed again, it would be better done so in a prospective study where a CCEAI score can be reliably produced for all cases.
Line 180: Comma after “effusion”
The authors would like to thank the reviewer for this comment.
Line 210 has now been altered to read as follows: ‘thickening of the intestinal mucosa, focal peritoneal effusion, and loss of wall layering.’
Line 188: Please explain the reason why these 3 cases underwent further IHC +/- PARR analysis, and why only these 3 cases? This should be a limitation discussed in Discussion.
The authors would like to thank the reviewer for these comments, LINE 214 has now been edited to read as follows:
‘Further analysis was performed in three cases at the time of diagnosis where the hematoxylin and eosin (H&E) histopathologic diagnosis had been classed as inflammatory but concern was held by the clinical pathologist that the cellular infiltrate was trending towards being predominantly lymphocytic. Two cases underwent immunohistochemistry (IHC) and one underwent both IHC and PARR analysis, confirming the presence of lymphoma.’
Line 223-224: Consider “Statistical significance was set at P < 0.05.” with a space after P
The authors would like to thank the reviewer for this comment, this has now been edited between LINES 211- 225, as it was noted that this was inconsistent throughout these paragraphs.
Line 259-261: Consider “biopsies” rather than “biopsy”, and remove “…of the duodenum and ileum” as that is already implied in the methods.
The authors would like to thank the reviewer for this comment, this has now been edited on LINE 261 to read as follows:
‘In total, 79 dogs were included in the study. Seventy-four underwent endoscopic biopsies and 5 underwent full-thickness biopsies.’
Line 259-261: Consider “biopsies” rather than “biopsy. Also, this sentence is confusing. So did 5 dogs have both endoscopic AND full-thickness biopsies, as that is what it sounds like with the word “concurrent”, or did you mean concurrent in the sense of having both duodenal and ileal biopsies? (assuming the latter is the case given your results / numbers, concurrent is redundant given the previous comment).
The authors would like to thank the reviewer for this comment, this has now been edited on LINE 262 to read as follows:
‘In total, 79 dogs were included in the study. Seventy-four underwent endoscopic biopsies and 5 underwent full-thickness biopsies.’
Line 337: Consider “The breeds within the endoscopy biopsy group included German Shepherd dog…” etc
The authors thank the reviewer for this comment, this has now been altered to read as follows (LINE 267):
‘The breeds within the endoscopic biopsy group included German Shepherd dog (10), Labrador (7), crossbreed (6), Rottweiler (5), Cocker Spaniel (4), and Staffordshire Bull Terrier (4).’
Line 339: Comma after (4)
The authors would like to thank the reviewer for this comment. This has now been altered within the manuscript. LINE 268 reads as follows:
‘Shepherd dog (10), Labrador (7), crossbreed (6), Rottweiler (5), Cocker Spaniel (4), and’
Line 240: Consider “The breeds within the full-thickness biopsy group included…” and list the breeds.
The authors thank the reviewer for this comment, this line has now been edited to read as follows: LINE 269.
‘The breeds within the full-thickness biopsy group included Dalmatian (1), Chihuahua (1), cross breed (1), Great Dane (1), Pug (1).’
Line 278: The fact that some dogs are missing signalment and BCS data, this should be stated at the beginning of the paragraph. Also, how many dogs were missing signalment data from each group should be stated, leaving how many dogs with complete signalment data in each group that was available in each group also stated before you start listing signalment data, particularly given that signalment is one of the results you are looking for association with discordance data.
The authors thank the reviewer for this comment.
The signalment data referred to as missing is the body condition score data for the 6/74 endoscopic cases. We apologise for any confusion caused and have now edited this to relay that only BCS data was missing.
LINE 252 now reads as follows:
‘Data for body condition score was not available for 6/74 endoscopic cases.’
Line 252-253: Consider “…clinical signs included diarrhea (69 dogs; characterized as large bowel [15], small bowel [15], mixed [39], and hemorrhagic [6]), vomiting (35 dogs; of which 6 were characterized as hemorrhagic)….” etc
The authors would like to thank the reviewer for this comment.
Line 282 has now been altered to read:
In the endoscopic biopsy group, clinical signs were as follows: diarrhea (69 dogs, characterised as large bowel [15], small bowel [15] and mixed bowel [39]), hemorrhagic diarrhoea (6), vomiting (35 dogs, of which 6 where hemorrhagic), lethargy (15), weight loss (31), reduced appetite (24), coprophagia (4), abdominal pain (4), melena (3), regurgitation (1), tenesmus (1), increased appetite (1), bloat (1), and borborygmi (1).
Line 258: Please elaborate on the lymphadenopathy – which ones / where?
The authors would like to thank the reviewer for this comment. This has now been altered to peripheral lymphadenopathy, as the lymph nodes listed encompassed prominent peripheral lymph nodes. The sentence now reads as the following (LINE 290):
‘In the full-thickness biopsy group, clinical signs included mixed bowel diarrhea (3), vomiting (4), regurgitation (2), weight loss (3), decreased appetite (2), peripheral lymphadenopathy (1) and bloat (1).’
Line 259: Comma after (1) The authors would like to thank the reviewer for this comment, this has now been altered on LINE 290, to read as follows:
‘In the full-thickness biopsy group, clinical signs included mixed bowel diarrhea (3), vomiting (4), regurgitation (2), weight loss (3), decreased appetite (2), peripheral lymphadenopathy (1), and bloat (1).’
Section 3.3: Why did you suddenly list full-thickness biopsy group’s data first when every other section, endoscopic biopsy group was listed first. Please be consistent.
The authors thank the reviewer for this comment. This has now been altered to read as follows, LINE 300:
In the endoscopic biopsy group, hypoalbuminemia (<28 g/L) was recorded in 27 of the 74 cases, hypoglobulinemia (< 21g/L) was detected in 17 of 74 cases, hypocobalaminemia (<200 ng/L) was noted in 33 of the 74 cases, and hypofolatemia (<7.1 ug/L) was identified in 18 of the 74 cases. Data was missing for cobalamin in 1 case and for folate in 4 cases. In the full-thickness biopsy group, hypoalbuminemia (<28 g/L) was seen in 2 of the 5 cases, hypocobalaminemia (<200 ng/L) was noted in 2 of the 5 cases and hypofolatemia (<7.1 ug/L) was identified in 1 of the 5 cases. Data for albumin and globulin was available for all cases, but absent for cobalamin in 2 cases and for folate in 2 cases.
Line 264-265: Listing hypocobalaminemia as 2/5 and hypofolatemia as 1/5 cases is misleading, given you are missing cobalamin and folate data in 2 dogs each. It should be 2/3 and 1/3 cases where the data was available. Need to rephrase the paragraph. State missing data before relevant results.
Line 270: Listing hypocobalaminemia as 33 of 74 cases and hypofolatemia as 18/74 cases is misleading, given you missing some cobalamin and folate data. It should be 33//73 and 18/70 cases where the data was available. State missing data before relevant results.
The authors would like to thank the reviewer for these comments. This paragraph now reads as follows (LINE 300):
‘In the endoscopic biopsy group, data was missing for cobalamin in 1 case and for folate in 4 cases. Hypoalbuminemia (<28 g/L) was recorded in 27 of the 74 cases, hypoglobulinemia (< 21g/L) was detected in 17 of 74 cases, hypocobalaminemia (<200 ng/L) was noted in 33 of the 73 cases, and hypofolatemia (<7.1 ug/L) was identified in 18 of the 70 cases.
In the full-thickness biopsy group, data for albumin and globulin was available for all cases, but absent for cobalamin in 2 cases and for folate in 2 cases. Hypoalbuminemia (<28 g/L) was seen in 2 of the 5 cases, hypocobalaminemia (<200 ng/L) was noted in 2 of the 5 cases and hypofolatemia (<7.1 ug/L) was identified in 1 of the 5 cases.’
Line 280: FNA results for the mesenteric lymphadenopathy casa available (esp. since that appears to be one of the lymphoma cases based on results elsewhere)?
The authors would like to thank the reviewer for this comment, our apologies if there is confusion but all lymphoma cases are based on intestinal biopsy samples. No mesenteric lymph nodes were sampled, and none of the diagnoses in this study were based upon lymph node results. The lymphadenopathy reported was a prominence of the peripheral lymph nodes.
Please list / summarise the available demographic data in dogs of each group (signalment, weight, BCS), clinical signs, clinicopathologic data, including all the CCEAI data and scores, and ultrasound data in a separate table for clarity, with numbers and percentage listed for cases with actual available data (ie. hypofolatemia.(18/70, 24%)
The authors thank the reviewer for this comment. As this information is present in the main body of the results in a clear way, a table was not included. However, if this is deemed integral, then the authors would be happy to create such a table.
Section 3.5 Final Diagnosis: Please make sure you discuss the changes in diagnosis after repeat H&E assessment in the discussion.
The authors thank the reviewer for this comment. The authors have added a paragraph regarding alterations in the diagnosis on LINE 475, which reads as follows:
‘In 7 of the CIE cases from this case population (5 from the endoscopy biopsy group and 2 from the full-thickness biopsy group) review of the slides for the purposes of this study altered their diagnosis from mild CIE to normal biopsy infiltrate. These slides were taken from older specimens within the biopsy pool and since the initial reporting the following guidelines have been produced; WSAVA Guidelines for evaluation of gastrointestinal inflammation in companion animals 2010 and the simplified scoring system for defining gastrointestinal Inflammation produced by Allenspach et al 2019 1,20. The use of these systems has improved the definition of histopathologic parameters and their grading, allowing more confidence in interpreting ‘normal’ biopsy infiltration.’
Line 284: Add “biopsy” after “endoscopic”. There have been several places in the manuscript where sometimes you put endoscopy or full-thickness biopsy group, and sometimes you miss the word “biopsy”. Please make sure they are consistent. If it is cumbersome and you want to abbreviate as EB and FTB groups, that is fine, but either way you need to make sure it is consistent throughout the manuscript.
Line 337: “…endoscopic biopsy group” instead of “endoscopy group”
The authors would like to thank the reviewer for this comment. Please note this has now been changed throughout the manuscript.
Lines 288, 303: Remove the infectious sentences.
The authors thank the reviewer for this comment. LINE 325, has now been altered to remove the mention of infectious cases.
In the endoscopic biopsy group, histopathologic diagnosis was of chronic inflammatory disease in 64 of 74 cases and EATL-2 lymphoma in 5 of 74 cases. In 5 cases that were originally diagnosed as CIE based on mild inflammatory infiltrate, histopathologic review for the purposes of this study deemed both the duodenal and ileal biopsy specimens to have a normal infiltrate (Table 1).
Table 2: I’m confused by the results. You said there are 3 CIE cases, but you have 1 concordant, and then you have a neutrophilic in the duodenum and eosinophilic in the ileum which I assume is 1 case, so that makes only 2 CIE cases. Where is the third case?
The authors thank the reviewer for this comment. The authors apologise for any confusion, the column for those cases with concordant diagnosis includes 4 cases- 2 CIE and 2 with normal biopsy infiltrate. The concordant column means that both cases were of that CIE type, so there is one lymphoplasmacytic enteritis and one lymphoplasmacytic and eosinophilic enteritis, which were concordant. There was then one discordant in histopathologic diagnosis by site, where the duodenum was neutrophilic and the ileum was eosinophilic and neutrophilic, the type of CIE is recorded at each site. Please let the authors know whether this remains confusing.
Table 3: Lymphoma EATL-2 should be included in the tables rather than just listing the enteritis diagnoses
The authors thank the reviewer for this comment. Lymphoma cases were not included, as this table was to demonstrate the inflammatory score of CIE only, as severity is not used for lymphoma it was not included.
Table 4: There should be space between 20/74 and cases
The authors would like to thank the reviewer for this comment. This table (Table 4) has now been edited to display correctly (LINE 373).
Line 376: Rephrase to “ Three cases in the endoscopic biopsy group…”
The authors would like to thank the reviewer for this comment. With the line numbers it was hard to isolate the correct sentence, the following line has been altered as we suspect this was the one the reviewer was referring to. LINE 389:
‘Three cases in the endoscopic biopsy group of EATL-2 were diagnosed in both the duodenal and ileal biopsy specimens.’
Line 376: Why was the decision made to perform further testing on these 3 cases but not others? And why some had IHC only and some had PARR as well?
The authors would like to thank the reviewer for this comment.
Unfortunately, as this was a retrospective study, dating from 2008 to 2019 and these tests were performed at the time of diagnosis, the reasons behind choosing IHC or PARR were not disclosed in the electronic referral hospital records.
Lines 377-379: Please list the IHC results (were they both CD3+ positive but CD20- ?)
The authors would like to thank the reviewer for this comment, please find the results listed within LINE 400.
‘IHC (CD3 and CD20) alone was performed in two cases; both were CD3 positive diagnosing EATL-2 in both the duodenum and ileum in one case and EATL-2 located solely within the duodenum in the other. An additional case diagnosed as marked lymphoplasmacytic inflammation underwent IHC (CD3 positive predominantly, CD79 weakly positive) and PARR analysis (B clonal lymphocyte population) and was found to be consistent with lymphoma (EATL-2) in both the duodenal and ileal locations.’
Line 381: Seems odd that you have CD3+ on IHC but then B clonal on PARR. How was the PARR performed? Capillary vs gel? Duplicate or better triplicate? Are you performing PARR based on your results from IHC or independently? Are you evaluating only TCRG for T cell or more in case the peak was obscured by a polyclonal background for T cell, or are you evaluating for kappa deleting elements or just IGH for B cell assay etc? As far as I understand, although cross-lineage expression can occur, it is rare for T cells to rearrange B cell loci. Lineage assignment should be based on immunophenotyping and not by clonality testing. If your sample was CD3+ but CD20, I suspect the result from immunophenotyping was more accurate given the limitations of PARR that is well referenced in human and veterinary literature. Given the discordance between IHC and PARR, this needs to be addressed in the discussion.
The authors would like to thank the reviewer for this comment. The reviewer has raised an interesting point, however, this is beyond the scope of this study, given the few number of cases that subsequently underwent IHC and PARR. However, the following has now been added to the discussion as recommended:
LINE 550:
‘One of the EATL-2 endoscopic biopsy cases underwent IHC and was strongly CD3 positive, but also demonstrated a weak positivity for CD79a, further to this PARR analysis identified a B-clonal population. PARR lacks sensitivity for canine intestinal lymphoma with sensitivities of only 66.7-75% being reported, this is speculated to be due to the significant level of inflammatory infiltration seen alongside neoplastic cells in cases with EATL-2 31,32. Its’ specificity for EATL-2 has also been brought into question in that 51% of CIE cases in a study on Shiba dogs demonstrated monoclonal populations 33. CD79a is a B-cell marker, co-expression of CD3 and CD79a has been identified in canine nodal lymphoma and co-expression of CD3 and CD20 has been documented in canine EATL-1 34,35. Given the weak expression of CD79a in the case from our study, and based on H&E assessment alongside strong staining for CD3 this case was diagnosed with EATL-2, however, the presence of co-expression and lack of sensitivity and specificity of PARR for EATL-2 highlights the need for cases to be assessed with combined techniques using H&E, IHC and PARR analysis.’
Lines 389-392:
Is the P value for folate 0.02 or 0.22? I assume 0.22? Please rephrase and with appropriate spaces:
“…biopsy type (endoscopic versus full thickness, P = 0.57), age (P = 0.70), sex and neuter status (P = 0.88), albumin (P = 0.23), globulin (P = 0.43), cobalamin (P = 0.53), folate (P = 0.22), and ultrasound findings (P = 0.73). Multivariable analysis was not performed as no variable had a P < 0.20.”
The authors would like to thank the reviewer for the above two comments. Please find that this paragraph has now been altered as follows, LINE 412.
‘No variables were found to have a statistically significant association with discordance of histopathologic diagnosis in the univariable analysis: biopsy type (endoscopic biopsy versus full-thickness biopsy) P = 0.57, age P = 0.70, sex and neuter status P = 0.88, albumin P = 0.23, globulin P = 0.43, cobalamin P = 0.53, folate P = 0.22, and ultrasound findings P = 0.73. Multivariable analysis was not performed as no variable had a P < 0.20.’
Discussion:
Line 407: Add “both” before inflammatory to make it clearer that this isn’t the first study evaluating agreement between locations, but yours also has lymphoma rather than just CIE
The authors thank the reviewer for this comment. LINE 422 has now been altered to read:
‘between duodenal and ileal biopsies in dogs with both inflammatory and neoplastic’
Line 398: Take out “different biopsy types:” and just say “…between endoscopy and full-thickness biopsies.”
The authors thank the reviewer for this comment. LINE 409 has now been altered to read:
‘etiologies and contrasting concordance rates between endoscopic biopsy and full-thickness biopsy’
Line 403-405: Equitable is a strange choice of word. Consider rephrasing as “Our study showed a reasonable agreement between endoscopic and full-thickness duodenal and ileal biopsy specimens, with moderate to substantial agreement identified, respectively.” etc
The authors would like to thank the reviewer for this comment. The authors have changed equitable to similar but have not re-phrased the sentence as when re-phrased it seemed to suggest that the endoscopic and full thickness specimens were taken from the same patient. We hope this alteration seems reasonable to the reviewer. LINE 428.
‘Our study showed that a similar level of concordance in histopathologic disease diagnosis exists between endoscopic and full-thickness duodenal and ileal biopsy specimens, with moderate to substantial agreement identified respectively.’
Line 405-408: Cumbersome sentence. Rephase: This was higher than the 27% agreement reported by Casamian Sorrosal et al in dogs with canine chronic enteropathy, although none of their cases were diagnosed with neoplasia.”
The authors would like to thank the reviewer for this comment. LINE 431 now reads as follows:
‘This was higher than the 27% agreement reported by Casamian-Sorrosal et al in dogs with canine chronic enteropathy, although none of their cases were diagnosed with neoplasia 3.’
Line 408-409: Be consistent with your tenses. Rephrase:
“We found higher rates of concordance when agreement between severity of disease was not evaluated, which was supported by findings from other literature.”
The authors would like to thank the reviewer for this comment. LINE 434 has now been edited as follows:
‘We found higher rates of concordance when agreement between severity of disease was not evaluated which is supported by findings from other literature 3,4.’
Lines 416-419: Why do you think your population had more duodenal abnormalities vs ileum, in contrast to the 2 other studies? Please discuss.
The authors would like to thank the reviewer for this comment. Please note we have added two additional sentences to discuss this: LINE 446.
‘This could be due to the duodenum being more accessible, which facilitates sampling along its’ length, increasing the likelihood of obtaining affected tissue. This is compared to the ileum where access is more difficult and so biopsies of a smaller region may be obtained. Additionally, a blind biopsy technique can be performed when access to the ileum cannot be achieved via endoscopy, this again increases the risk of missing lesions especially if the ileal mucosa cannot be visually inspected.’
Line 421: I don’t understand why you are bringing up biopsy in this discussion of more duodenal abnormalities vs ileal? Your study excluded samples with poor quality. Also, I’d like you to include a small section about biopsy quality in your results – were they all good / adequate, or were any marginal but also included in the study. And the proportional of those in duodenal vs ileal samples, whether or not they had any association with the level of discordance, since biopsy quality may influence diagnosis.
The authors thank the reviewer for this comment. Biopsy quality was not discussed as only those slides of adequate quality were accepted due to the retrospective nature of the study. There was, therefore, no avenue to explore regarding biopsy quality versus discordance.
For clarity the authors have now amended the section within the methods were this is discussed, LINE 173, this now reads:
‘At this stage cases were excluded if the quality of the histopathologic sections was deemed inadequate or marginal due to incomplete length of villi and depth of crypts or if slides were no longer available for review.’
Line 453: The issue isn’t as much that the EATL-2 cases were not correctly classified (other then that case with discordant IHC / PARR results which should be discussed), but rather the main limitation is that there is a possibility some EATL-2 cases may have been missed with only had H&E assessment.
The authors thank the reviewer for this comment. This sentence has now been altered to read: LINE 499
‘that not all EATL-2 cases were captured based on H&E assessment alone.’
Lines 422-424: Cumbersome sentence. Remove “the” with “our” cases, remove “included in our study”, insert comma after study, replace “and as a result” with “therefore”; replace “these inflammatory disease cases” with “our CIE cases”. If you are going to establish abbreviations, you should perhaps use them.
The authors thank the reviewer for this comment. This sentence has now been altered to read as follows: LINE 454.
‘Unfortunately, outcome data was not available for our cases therefore, we could not evaluate the impact of histopathologic discordance on our CIE cases.’
Line 426: Remove “clear”
The authors thank the reviewer for this comment. This sentence has now been altered to read: LINE 458
‘Numerous studies have failed to demonstrate associations between’
Line 428: Remove that ] after the reference number
The authors thank the reviewer for this comment. This sentence has now been altered to read: LINE 460
‘outcome 22–25.’
Lines 436-437: Would like you to also evaluate CCEAI and its association.
Also, rephrase as: “The lack of association between clinicopathologic variable and the level of histopathologic discordance between duodenum and ileum in our study, and the lack of understanding regarding the clinical impact of such histopathologic discordance, suggests that concurrent duodenal and ileal biopsy samples should be ideally obtained where possible.”
The authors thank the reviewer for this comment. The paragraph has now been altered as follows (LINE 469):
“The lack of association between clinicopathologic variable and the level of histopathologic discordance between duodenum and ileum in our study, and the lack of understanding regarding the clinical impact of such histopathologic discordance, suggests that concurrent duodenal and ileal biopsy samples should be ideally obtained where possible.”
The retrospective nature of this study and the wide time span over which the study was conducted makes CCEAI difficult to establish reliably for all cases from the electronic medical records.
Line 442: Why do you think all your lymphoma cases were EATL-2. Was it because EATL-1 were less likely to have undergone endoscopy based on ultrasound more mass-like findings? Or other reasons? Discuss.
The authors thank the reviewer for this comment; this has been discussed within the limitations section of the study as follows. LINE 585.
‘Further to this, our study cohort did not contain any canines diagnosed with EATL-1, the more common alimentary lymphoma in dogs. This is likely to have occurred due to cases with EATL-1 being more readily identified from ultrasonographic abnormalities and then diagnosed from cytology of fine needle aspirates or histopathology of surgical biopsies that were lesion targeted and therefore did not match the criteria for case selection.’
Lines 444-445: I am confused by your comparison. Your study looked at agreement, the Couto study did not. They had 85% LSA in the duodenum but only 54% in the ileum, and did not look at level of discordance. Please rewrite this statement as I do not understand the point you are trying to make here. The comparison with Couto’s study may be more appropriate later in the paragraph when you discuss your 40% EATL-2 in the duodenum.
The authors thank the reviewer for this comment. This has now been altered accordingly: LINE 500.
‘The predilection of EATL-2 cases in our study was for the duodenum, this was similar to a recent study where 85% of EATL-2 cases occurred at the duodenum compared to only 54% in the ileum 12. This is in contrast to feline EATL-2, which has a predilection for the ileum and jejunum 27.’
Line 454-455: See comments for Line 72 re EATL-2 distribution in cats. Please include those references and discuss further.
The authors would like to thank the reviewer for this comment. LINE 502 now reads as follows:
‘This is in contrast to the findings of a number of studies on feline EATL-2, where the jejunum and ileum are reported to be the favoured locations, however; another study has found the majority of cases occurred in the duodenum and jejunum 10,13–15,32.’
The authors have added this to the general discussion on favoured locations as follows, LINES 500:
‘The predilection of EATL-2 cases in our study was for the duodenum, this was similar to a recent study where 85% of EATL-2 cases occurred at the duodenum compared to only 54% in the ileum 12. This is in contrast to the findings of a number of studies on feline EATL-2, where the jejunum and ileum are reported as the favoured locations, however another study found the majority of cases occurred in the duodenum and jejunum10,13–15,32. Forty percent of the EATL-2 cases in our study occurred in the duodenum alone with lymphoplasmacytic and eosinophilic inflammatory disease identified in the ileum. In those cases assessed via IHC or IHC and PARR, EATL-2 was detected in both the duodenum and ileum or in the duodenum alone, and in none of these cases would sampling of the ileum have altered the diagnosis already obtained in sampling the duodenum. This is similar to recent findings in feline intestinal lymphoma, where sampling of the ileum rarely changed diagnosis when biopsies were assessed with a combination of H&E staining, IHC and PARR 16.’
Line 459: Again, similar to previous comment for Lines 75-76, I don’t think you can say the Lane et al study suggested preferential EATL-2 in the ileum.
The authors would like to thank the reviewer for this comment. LINE 518 has now been edited to read as follows:
‘Clinically, this could lead to prioritization of upper GI endoscopy where EATL-2 is suspected, but another study has found EATL-2 occurring at the ileum in many dogs, including detection in the ileum alone’
Line 462-466:
Please discuss more regarding why you think more EATL-2 was noted in the EB group rather than FTB group. Perhaps, because you had a bigger EB group, or perhaps as previously mentioned in another comment, FTBs may not in fact be superior to EBs for EATL-2 due to less diagnostic surface area from FTBs? Or any other reaons?
The authors would like to thank the reviewer for this comment. LINE 521 have now been altered to read as follows:
‘Interestingly, the neoplasia cases in our study existed solely within the endoscopic biopsy group. Whilst in large part this could be due to the larger number of cases within this biopsy group, feline EATL-2 has been shown to manifest initially in the mucosal layer for which endoscopic biopsy sampling may be better suited than full-thickness biopsy 13,15,16. Full-thickness biopsy has been demonstrated to provide comparatively smaller samples of mucosa and a lower number of samples at each site are procured 23. With evidence demonstrating that ileal biopsy rarely impacted diagnosis in cases of feline EATL-2 assessed with H&E, IHC and PARR and the possibility of reaching the proximal jejunum in small dogs with gastroduodenoscopy, this calls into question the need to use full-thickness biopsy in these cases unless a lesion has been noted at the mid jejunum 16,21. Further research is required on the need for concurrent ileal biopsy in dogs with suspected EATL-2 if IHC and PARR analysis are also used before any definitive conclusions regarding the requirement for duodenal and ileal biopsy can be made.’
Also, some of your citations do not support your point. Scott et al. study looked at only endoscopic biopsies and did not compare with FTBs. Lane et al discussed if FTB may have been suggested as superior to EBs in cats (but did not actually evaluate that in their canine study). Also, based on what we know we cats, although certain features from FTBs may help diagnose LSA, alimentary SCLSA in cats manifests in the mucosa long before transmural progression, as discussed by Moore et al, Carerras et al, Fondacaro et al, and most recently Chow et al that evaluated the agreement between feline endoscopic biopsies in the duodenum and ileum.
Also, although Evans et al study suggest FTB may be superior, they did not biopsy ileum endoscopically compared to FTBs, and some of the endoscopic duodenal biopsies were obtained blindly, which may affect biopsy quality and also diagnostic surface area compared to cannulating with direct visualization.
All these should be discussed as your conclusion / statements are very misleading right now, and your citations do not actually support what you said.
The authors would like to thank the reviewer for this comment, and apologise for the misleading information. LINES 521 onwards now read as follows:
‘Interestingly, the neoplasia cases in our study existed solely within the endoscopic biopsy group. Whilst in large part this could be due to the larger number of cases within this biopsy group, feline EATL-2 has been shown to manifest initially in the mucosal layer for which endoscopic biopsy sampling may be better suited than full-thickness biopsy 13,15,16. Full-thickness biopsy has been demonstrated to provide comparatively smaller samples of mucosa and a lower number of samples at each site are procured 23.. With evidence demonstrating that ileal biopsy rarely impacted diagnosis in cases of feline EATL-2 assessed with H&E, IHC and PARR and the possibility of reaching the proximal jejunum in small dogs with gastroduodenoscopy, this calls into question the need to use full-thickness biopsy in these cases unless a lesion has been noted at the mid jejunum 16,21. Further research is required on the need for concurrent ileal biopsy in dogs with suspected EATL-2 if IHC and PARR analysis are also used before any definitive conclusions regarding the requirement for duodenal and ileal biopsy can be made.’
Limitations:
Line 491: Add “biopsy” after full-thickness
The authors would like to thank the reviewer for this comment. This line has now been altered to read as follows: LINE 524.
‘population and as such this led to more exclusions being made within the full-thickness biopsy group.’
Line 494: Again, remove “infectious” as that has no relevance in this paper since you had no infectious cases.
Rephrase as “…concordance of inflammatory and neoplastic enteropathies. In addition, the concordance of various morphologic criteria of CIE between intestinal locations has been previously evaluated by Procoli et al 2013.”
The authors would like to thank the reviewer for this comment. LINES 578 have now been edited as follows:
‘specifically aimed to assess the concordance of inflammatory and neoplastic enteropathies. In addition, the concordance of various morphologic criteria of CIE between intestinal locations has been evaluated previously by Procoli et al 2013.’
Line 496-498: Rephrase “Although our study initially aimed to also evaluate infectious cases, none of the dogs in our study population was diagnosed with an infectious chronic enteropathy.”
The authors would like to thank the reviewer for this comment. LINE 580 has now been removed as we have removed any mention of the inclusion of infectious cases within the manuscript as its’ presence may be confusing and is also irrelevant to the manuscript. The following paragraph has been deleted:
‘Although our study aimed to include all GI etiologies i.e. inflammatory, neoplastic, and infectious, the case population did not include any infectious cases. This is likely due to the geographic location of where the study was conducted, where fungal diseases were not endemic, and may also relate to antibiotic therapy trials being carried out in first opinion practice prior to cases reaching a referral setting, resulting in a skewed disease population. Further to this,’
Line 499: Replace “are” with “were” ; insert comma after “endemic”
The authors would like to thank the reviewer for this comment. LINE 533 has now been edited as follows:
‘study was conducted, where fungal diseases were not endemic, and may also relate to’
Line 501: Insert comma after “setting”
The authors would like to thank the reviewer for this comment. LINE 535 has now been edited as follows:
‘referral setting, resulting in a skewed disease population.’
Line 502: Replace “contain” with “include”; replace “canines” with “dogs”
The authors would like to thank the reviewer for this comment. LINE 589 has now been edited as follows:
‘did not include any dogs diagnosed with EATL-1, the more common’
Line 504: Replace “from cytology of fine needle aspirates” with “cytologically by fine needle aspirates”
Line 505: Insert comma after “aspirates”
The authors would like to thank the reviewer for these comments. LINE 589 has now been edited as follows:
‘cytologically by fine needle aspirates, or by histopathology of surgical’
Only some cases had IHC +/- PARR – discuss the limitations but also touch on the issues associated with clonality test in regards to sensitivity / specificity, as also brought up by your discordant IHC / clonality result as I have previously commented on. Keller et al, Moore et al, Marsilio et al, and Chow et al may be good references for you to cite and read through for a better understanding if needed.
The authors would like to thank the reviewer for this comment; a new paragraph discussing the IHC and PARR analysis, and sensitivity and specificity has been added.
LINE 550.
‘One of the EATL-2 endoscopic biopsy cases underwent IHC and was strongly CD3 positive, but also demonstrated a weak positivity for CD79a, further to this PARR analysis identified a B-clonal population. PARR lacks sensitivity for canine intestinal lymphoma with sensitivities of only 66.7-75% being reported, this is speculated to be due to the significant level of inflammatory infiltration seen alongside neoplastic cells in cases with EATL-2 31,32. Its’ specificity for EATL-2 has also been brought into question in that 51% of CIE cases in a study on Shiba dogs demonstrated monoclonal populations 33. CD79a is a B-cell marker, co-expression of CD3 and CD79a has been identified in canine nodal lymphoma and co-expression of CD3 and CD20 has been documented in canine EATL-1 34,35. Given the weak expression of CD79a in the case from our study, and based on H&E assessment alongside strong staining for CD3 this case was diagnosed with EATL-2, however, the presence of co-expression and lack of sensitivity and specificity of PARR for EATL-2 highlights the need for cases to be assessed with combined techniques using H&E, IHC and PARR analysis.’
Conclusions:
Line 509: Consider rephrase: “In conclusion, there were similar rates of histopathologic concordance and discordance for duodenal and ileal biopsies between endoscopic and full-thickness biopsy specimens. Canine EATL-2 demonstrated a predilection for the duodenum in our case population. Until we better understand the clinical impact of histopathologic discordance, concurrent biopsy of the duodenum and ileum may be ideal in cases suspected to have CIE. Further research with a large case population and an improved understanding of the clinical impact of histopathologic discordance between duodenum and ileum are needed.”
The authors would like to thank the reviewer for this comment. The authors agree that this is improved phrasing and have inserted this into the manuscript as written above. This can be found on LINE 593.